# The Virtual Navigation Toolbox: Providing tools for virtual navigation experiments

**Martin M. Müller**[1] *, **Jonas Scherer**[1], **Patrick Unterbrink**[1], **Olivier J. N. Bertrand**[1], **Martin Egelhaaf**[1], **Norbert Boeddeker**[2]

**1** Department of Neurobiology, Bielefeld University, Bielefeld, NRW, Germany, **2** Department of Cognitive Neuroscience, Bielefeld University, Bielefeld, NRW, Germany

* martin.mueller@uni-bielefeld.de

**Data Availability Statement:** The data underlying the results presented in the study, as well as the source code and compiled versions of the presented software tools and showcase

## Abstract

Spatial navigation research in humans increasingly relies on experiments using virtual reality (VR) tools, which allow for the creation of highly flexible, and immersive study environments, that can react to participant interaction in real time. Despite the popularity of VR, tools simplifying the creation and data management of such experiments are rare and often restricted to a specific scope—limiting usability and comparability. To overcome those limitations, we introduce the Virtual Navigation Toolbox (VNT), a collection of interchangeable and independent tools for the development of spatial navigation VR experiments using the popular Unity game engine. The VNT's features are packaged in loosely coupled and reusable modules, facilitating convenient implementation of diverse experimental designs. Here, we depict how the VNT fulfils feature requirements of different VR environments and experiments, guiding through the implementation and execution of a showcase study using the toolbox. The presented showcase study reveals that homing performance in a classic triangle completion task is invariant to translation velocity of the participant's avatar, but highly sensitive to the number of landmarks. The VNT is freely available under a creative commons license, and we invite researchers to contribute, extending and improving tools using the provided repository.

## 1. Introduction: Opportunities and challenges of virtual reality for spatial cognition

Spatial navigation research aims to find out how humans and other animals solve spatial tasks by using various external sources of information, like landmarks [1, 2], panoramic views, or object constellations [3] as well as proprioceptive information about their orientation [4] and self-motion [5]. The use of virtual reality (VR) tools is of particular interest for analysing how humans solve spatial tasks, such as recognising locations, establishing, learning and following routes, or returning to a previously visited location ("homing").

### 1.1. VR: The "new" frontier in spatial cognition

Rapid advances in computer and display technology have enabled the real-time presentation of dynamic, interactive, simulated environments (now usually referred to under the umbrella

experiment are available from https://gitlab.ub.uni-bielefeld.de/virtual_navigation_tools/.

**Funding:** This work was funded by the Deutsche Forschungsgemeinschaft (DFG) grant 460373158 (https://gepris.dfg.de/gepris/projekt/460373158). OJNB, NB and ME received the funding, MMM, JS and PU were funded as part of the project. We acknowledge the financial support of the Deutsche Forschungsgemeinschaft (DFG) and the Open Access Publication Fund of Bielefeld University for the article processing charge. The funders had no role in study design, data collection and analysis, decision to publish, or preparation of the manuscript.

**Competing interests:** The authors have declared that no competing interests exist.

term "VR" in the research context), which has revolutionised the investigation of perception, navigation, and cognition over the past three decades [6–8]. VR-tools have opened up a wide range of new possibilities for navigation research because they allow for precise control and rapid change of environmental conditions, which define a navigational task. This enables interactions in ecologically relevant and visually complex environments, combined with accurate tracking of spatial behaviour [7]. Virtual reality can even go beyond what is possible in real-world experiments, teleporting users [9, 10] or creating non-Euclidean spaces and other impossible spatial constructs, which can generate exciting new insights into the inner workings of human spatial cognition [11, 12]. Furthermore, VR experiments can easily be combined with other technologies like motion or eye tracking, providing multi-dimensional behavioural measures with high temporal and spatial precision [13, 14].

In the last years, 3D graphics engines have increasingly enabled researchers to create complex and interactive virtual worlds, which are required to perform research in virtual environments, 'in-house' without the aid of external programmers and designers (see for example [1, 5, 11, 15–17]).

## 1.2. Challenges of working with VR

Nevertheless, designing and executing spatial navigation tasks in the virtual realm still comes with a significant cost in development time. Even when using modern 3D game development software, like the popular and free to use Unity engine, many requirements need to be fulfilled to successfully run experiments in VR. These start with the choice of a presentation medium (desktop PC, head-mounted displays, etc.) and continue with systems to manage the structure of the experiment and present the necessary spatial and interactive features.

Over the last years, several software packages have been published within the research community, which aim to facilitate different aspects of this process of creating and running VR experiments.

To name just a few examples, the **bmlTUX** [18] package offers ui-driven tools for combinatorics, counterbalancing and randomization of trials. The **Landmarks** [19] package provides easy-to-use, drag-and-drop setup for a number of virtual experiments, with a focus on integration of head-mounted displays (HMD). The **USE** [20] provides powerful tools for interfacing with external hardware, as well as tools for state-based stimulus control, while the **UXF** [21] focuses on management of experimental sessions by providing logging and tracking features, integrated with an intuitive session-block-trial structure.

These tools are extremely useful when employed within their designated field of application, and often provide a good degree of flexibility by being usable for different task variants even beyond their originally intended scope.

However, the modularity they emphasise acts on the level of swapping components within their overall framework. They do not implement modularity on a structural level by focusing on re-usability of their constituent components outside the framework.

This means that while it might be possible, or indeed encouraged, to tweak the existing composition of those tools to serve a novel, but related, function (like for example swapping out task prefabs within the Landmarks [19] package to switch from a navigation to a pointing task), being able to extract the underlying functional components (i.e. collections of scripts and related assets) from within the available packages, while sometimes possible, clearly does not constitute the main design goal of these packages.

This can be problematic, since the design of experimental paradigms necessarily requires a high degree of adaptation and turnover of functional components, which can often exceed the degree to which existing packages can be modified while still retaining their intended benefit

of providing a clear structure and easy modes of interaction for the experimenter (like the drag-and-drop exchange of task modules in the landmarks package mentioned above).

This exposes a fundamental problem in software design, where trying to provide as much ease of use as possible to the end-user has to be traded off against flexibility and ease of adaptation. We posit that there is no fundamentally right or wrong answer to this problem, as different approaches favour different kinds of users and use cases, with packages allowing for fixed but clearly defined interactions providing greater benefits to more casual users or highly static use cases, while more flexible architectures can provide greater long-term re-usability at the cost of longer initial periods of familiarisation with the framework and its implementation for users.

The existing software packages fall on different points on this spectrum of ease of use vs. ease of adaptation, covering very different use cases and starting from very different design requirements. However, we feel that, to date, no available package has explicitly taken up ease of adaptation as its core principle.

This means that researchers trying to implement new experimental paradigms in the field of VR-driven spatial cognition, who could profit from re-using many different functional software components, which exist within and across the various available tools, either have to painstakingly extract those features from within the respective package, often against the more holistic design philosophy of that given package, or fully re-implement them on their own. This leads to instances of "re-inventing the wheel" for many software components which could be reused across tasks and experimental designs.

This limitation of existing packages is understandable, since software development work usually forms only a small part of scientific project design, leading to strong constraints on the time that can be invested into developing a given project. However, it remains clear that the spatial research community could profit from a shared platform, where researchers can submit their solutions to common problems in the design of VR experiments, to lower the collective burdens of design work needed to run immersive and dynamic experiments.

To provide such a platform, we have created the Virtual Navigation Toolbox (VNT). Below we will introduce the VNT and its features using the example of a showcase virtual navigation experiment implemented with the help of our toolbox.

## 2. The Virtual Navigation Toolbox: A modular library of VR tools

We have developed a software toolbox that provides core functional components to implement, run and analyse various virtual navigation experiments. In developing this toolbox, we focused on the modularity and ease of adaptation of its individual components so that researchers can use and re-use these individual components without having to design their projects around the overarching architecture of our tools. The result of this work is VNT. The VNT is developed for the free to use Unity game engine (version 2021.3 LTS and newer) and is freely available for download under GNU General Public License v3.0 (see section 'Toolbox Availability').

In this paper, we will introduce the VNT and its features, by showcasing one of the virtual experiments we conducted with the help of the VNT. We will use this showcase to guide the reader through the process of identifying the required features of a VR experiment and implementing them in a modularised manner using the VNT.

Furthermore, with this showcase, we highlight how our toolbox interacts with existing tools, by showing how we integrated our work with a popular pre-existing research tool for Unity (the Unity Experiment Framework or UXF, see section 'Integration with existing tools' and Fig 2), without sacrificing modularity. We also provide the project files for the showcase

study as a ready-to-use experiment for other people in the community who want to run virtual homing experiments.

## 3. Showcasing the VNT

To showcase our toolbox, we have carried out a well-known type of homing experiment, called the triangle completion task in a set of different virtual environments. This showcase experiment is intended to serve as an introduction for other researchers on how to use the VNT for the design and implementation of virtual spatial tasks.

In the following paragraphs, we will first introduce the specifics of the experiment to be implemented and then outline how the different modules of the VNT, like dynamic waypoint and world-creation systems, which allow for an easy way of changing environments on the fly, can be used in practice for implementing a virtual navigation experiment. The showcase experiment also shows how the VNT can be integrated with other frameworks, like the UXF, which we used for its powerful data logging capabilities and its intuitive session–block–trial architecture.

### 3.1. Triangle completion in virtual reality

The triangle completion task and its derivatives are a staple of navigation research and have been used to assess spatial memory and egocentric navigation abilities ever since the task was introduced by Loomis and colleagues in their landmark study of path integration ability in blind and sighted individuals three decades ago [22, 23]. In such a task, the participant is typically guided along two sides of a triangle, either actively or passively, and at the end of this path they are asked to point towards, or actively return to, their starting location [24–26]. The researcher then analyses the observed errors in homing or pointing and can compare error measures obtained in different environments (e.g. with or without landmarks [2], longer or shorter paths, or different turning angles [27, 28]) or from different groups (e.g. healthy individuals vs. those suffering from vestibular disorders [29]) to gain insights into the way these factors might have influenced the participant's behaviour.

The triangle completion task provides a straightforward way to investigate homing accuracy and precision and has therefore been used both for real-world, as well as virtual navigation experiments. While recent results indicate that translation of results between real-world and virtual versions of the task should be undertaken with caution [30], the triangle completion task continues to be a paradigm of choice for a broad range of spatial cognition studies, especially in the virtual realm.

### 3.2. Design goals for the VNT showcase

We have designed a virtual homing task in which participants performed different variants of a triangle completion task in a set of different virtual environments. Specifically, the showcase aims to examine the impact of different translation velocities, as well as the presence or absence of landmark objects on homing and pointing performance. Our study design is similar to one used by Jetzschke et al. [31], in which participants had to execute a triangle completion task in environments containing one to three identical landmarks surrounding the goal location. We aimed to replicate their design while adding the additional component of an embedded marker placement task. This addition allowed us to not only probe our participants'spatial knowledge at the end of their return trip, but also at different times during the homing process and, thus, at different locations in the environment.

The experiment was run in "desktop-VR" on the participant's own home computer, highlighting the potential of distributed remote experiments for navigation research. However,

due to the focus on modularity and ease of adaptation of the VNT, the showcase can easily be adapted to use different presentation media. To illustrate this, we created a version of the showcase experiment configured to use an HTC Vive Pro HMD that is also available in the repository accompanying this study (see section 'Toolbox Availability').

In the following paragraphs, we describe the specifics of the showcase study and its research question to formulate software requirements and how they are fulfilled by the VNT.

This kind of requirement analysis can be applied to any experiment involving software development and is useful to judge the effort and time required to implement a desired experimental design before starting a research project. In Table 1 we provided a short overview of requirement analyses for the showcase study described below, as well as for other experimental tasks currently being carried out in our group.

**3.2.1. Novel marker placement task.** Pointing at a goal is a common task in many spatial orientation experiments (cf. [32] for a classic application or [33] for a more recent example). However, previous versions of this task are limited to a directional estimate, disregarding distance. Since both distance and direction information can be of great interest in different spatial cognition tasks, we have implemented a 3D version of this task, in which participants are asked to place a marker directly in the 3D environment at the spot they believe their goal location to be. We call this novel variant of the pointing task a 'marker placement task'. In the showcase study, participants performed the marker placement task by moving around a marker object in the virtual environment using the mouse cursor, however, due to the modular nature of the VNT, the task can be easily adapted to use different input media. For a more detailed discussion of adapting components of the VNT, see the 'Example of interface driven design' section below.

**3.2.2 Showcase Part A: Homing at different speeds.** The participants performed a triangle completion task in a virtual steppe environment containing no additional landmark cues.

**Table 1. Examples of feature requirements for VR tasks.** Left column names example tasks, middle column lists features which need to be implemented to run the given task, right column names software components which can fulfil the respective requirement. Trap-lining refers to a type of route optimisation task where navigators have to find and collect resources distributed around an environment [34, 35]. Eye tracking refers to the inclusion of 3D gaze recording in other virtual spatial tasks, see also section "Example of interface driven design" for a discussion on implementing gaze-based ray-casting within the VNT. VNT modules are designed to be highly modular and re-usable, which is shown by the fact that the same modules can fulfil requirements across different experiments. Note for example that while data logging is provided by the UXF for our showcase triangle completion study, we also provide our own data logging functionalities in the Util module of the VNT and use it for data handling in an ongoing eye tracking study.

| Task | Requirement | Implementation |
|---|---|---|
| Triangle Completion (Showcase Study) | Manage session structure | UXF |
| | Manage flow of trials | VNT Trial State Machine |
| | Generate environments based on settings | VNT Tile-based Level Generator |
| | Guide player during outbound path | VNT Waypoint-based Player Guidance |
| | Enable pointing at goal location | VNT Pointing and Highlighting |
| | Log result data | UXF |
| Trap-lining | Manage session structure | UXF |
| | Detect visited nodes | VNT Waypoint-based Player Guidance |
| | Manage flow of trial | VNT Trial State Machine |
| | Log result data | UXF |
| Eye tracking | Provide Gaze Data | Pupil Labs |
| | Manage application flow | VNT Trial State Machine |
| | Visualise Gaze Data | VNT Pointing and Highlighting |
| | Log Gaze Data to file | VNT Utilities |

During this task, participants were first guided along two legs of a triangular path and were then asked to place a marker in the virtual environment where they estimated their goal location to be and then walk there. The marker placement task was repeated after they had reached a distance of 12.5 meters from the start of the return trip (see Fig 1) and the marker was always removed before participants were ale to continue homing.

In a series of trials, the translation velocity of the participant's avatar was modified between 1 meter per second (very slow walk) and 7.5 meters per second (sprint). This changes the magnitude of the translational optic flow received by the participant, which in turn might influence visually driven path integration.

Each condition was presented between 17 and 24 times in randomised order to $N = 8$ participants. For more details of the experimental procedure we refer the reader to the section 'Illustrating VNT architecture using the showcase study' below.

We recorded position errors across these different translation velocities to investigate the effect of walking pace on PI-based pointing and homing performance (see Fig 1).

**3.2.3. Showcase Part B: Homing with landmarks.** To investigate the impact of landmark count on pointing and homing performance, another set of $N = 10$ participants performed the same triangle completion task as in A but with the translation velocity set to an intermediate value of 2.5 meters per second. In contrast to experiment A, the environments in this experiment contained between one and three trees which could be used as landmark cues (see Fig 1). Conditions were again presented in randomised order for between 18 and 24 repetitions per condition.

**3.2.4. Ethics statement.** All experiments were conducted in accordance with the guidelines of the Deutsche Gesellschaft für Psychologie e.V. (DGPs) and approved by the Bielefeld University Ethics Committee. Participants were recruited and experiments were carried out during August of 2020. Age of participants at time of study ranged from 18 to 27 years. All participants gave written informed consent before participation and were compensated with a payment of 7 euros per hour. Authors did not have access to information that could identify individual participants during or after data collection.

## 3.3. Formulating the requirements: How to implement the showcase study

Knowing the specifics of the task to be implemented, we can pinpoint the functionalities required to run the planned showcase study and place them in a broader context of navigation experiments. To implement a VR-driven navigation task, we must first decide on which tools to use. For the purpose of the VNT and the showcase study, we have chosen the Unity game engine to create and present the virtual environments and execute the different steps of the experiment, which in the case of the Unity engine must be written in the form of C# code. To execute the planned triangle completion and marker placement tasks, we must have access to implementations of:

- a system of automatically presenting the different experimental conditions to the participant in a given order ("session management"),

- tools for managing the programmatic flow during an individual trial ("trial flow"),

- a tool for creating the specific virtual environment required for a given experimental condition ("level generator"),

- a method of guiding participants along the outbound legs of the intended path through the created environment ("player guidance"),

- tools to provide the necessary user interactions with the environment (e.g. "pointing"),

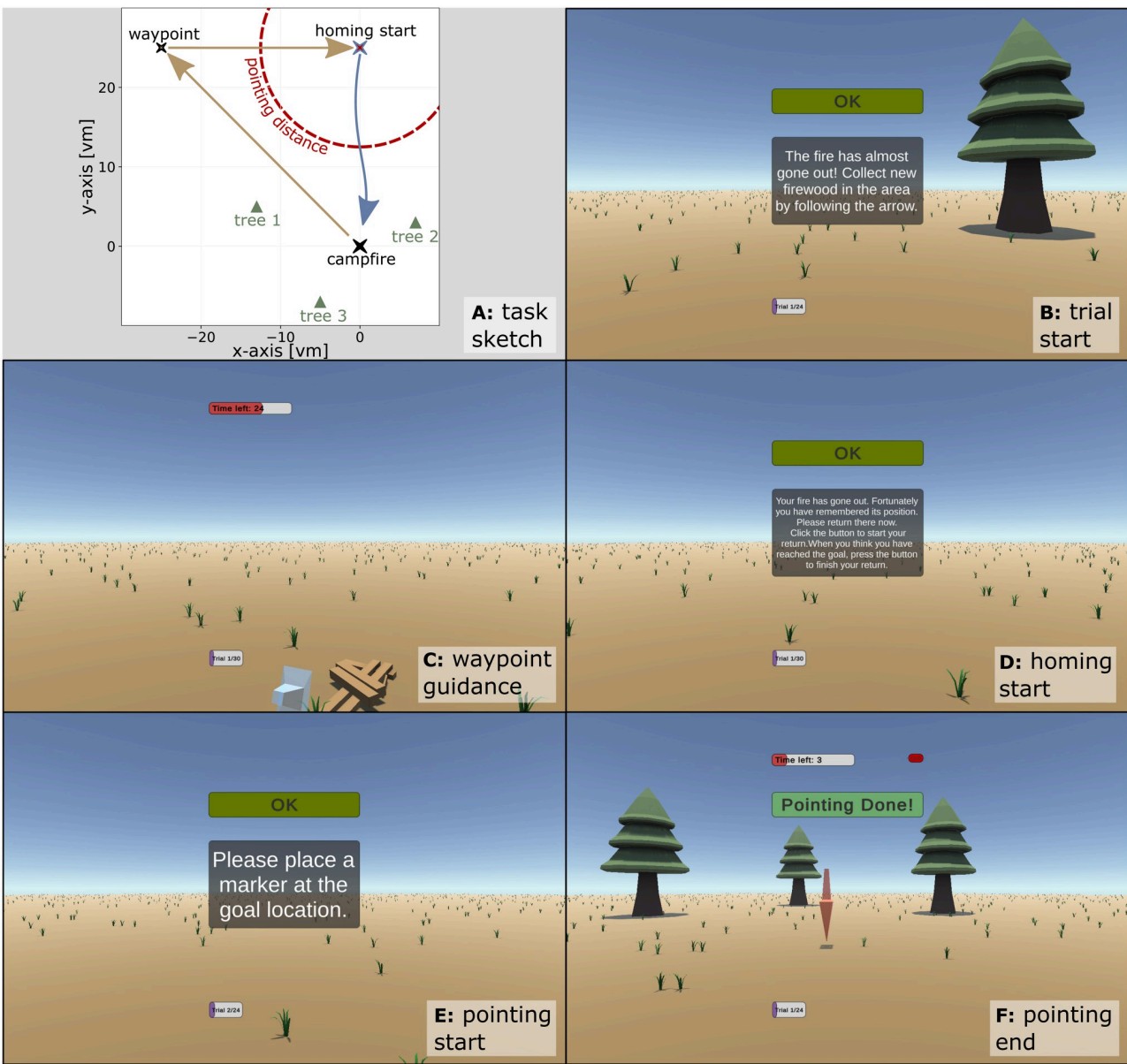

**Fig 1. Implementation showcase.** Subplot **A** shows the general layout of the implemented triangle completion task: Participants are guided along two legs of a triangle (beige arrows) and are then asked to first point at and then return to their starting location (a campfire). When reaching a distance of 12.5 meters from homing start (red dashed circle) participants are asked to point at their starting location a second time. The campfire was not visible during the pointing or return phase of each trial. In the first part of the showcase study, no trees were present and the translation velocity of the player avatar was manipulated, while in the second part of the study, 1, 2, or 3 trees were presented as shown in subplot **A** to aid in navigation, while the player speed was held constant at 2.5 *m/s*. These task variants were carried out repeatedly across several trials, either in the shown spatial arrangement, or in a design which mirrored the spatial layout along the world y-axis, yielding trials with either left or right-handed turns, which were presented in randomised order. Subplots **B-F** show the progression of an individual trial from the participant's perspective: Subplot **B** shows the instructions given at the start of the trial, subplot **C** shows the participant reaching one of the waypoints during the outbound phase of the trial (waypoints are pieces of firewood, which the participant needs to collect), subplot **D** shows the instructions given at the start of the return or homing phase, while subplots **E** and **F** show the instructions given for pointing and the resulting marker placement by the participant respectively.

- capturing and saving behavioural responses (walking and pointing) during the experiment ("data logging").

In Table 1, we have summarised the requirements for the showcase study and provide two more examples of experiments that we are currently conducting using the VNT and which we will make available as further showcase projects after they are concluded. These additional examples illustrate how our modular approach facilitates code reuse since many of the requirements exist across multiple experiments. Thus, by implementing them as encapsulated features ("tools in the toolbox"), we can use them again for other related tasks, such as the listed route optimisation or eye tracking experiments. Below, we will go over the specific modules which implement and fulfil the requirements of the showcase study.

### 3.4. Illustrating VNT architecture using the showcase study

To introduce the different modules of the VNT and how they fulfil the requirements listed above, we describe a single trial of the showcase study from the perspective of the participant and connect their experience during different parts of the trial to the implemented feature in question. To illustrate the "walk-through" of the trial described below, we refer to Fig 1 which shows the progression of the trial as seen from a participant's perspective, as well as Figs 2 and 3 which show the technical side of the progression along the trial.

At the start of each trial, the participant is guided to the starting position (a campfire) via an arrow, which is pointing at it. This is called the "Trial Start" phase and is encoded as a state in the Trial State Machine module of the VNT, which tracks the progress of the participant throughout the trial and triggers events and interactions as needed (see Fig 3). During this phase, the state machine causes the creation of the visual environment using the Tile-based Level Generator module of the VNT and then activates the Waypoint-based Player Guidance module of the VNT to guide the player to the starting position.

Once the participant has arrived there (controlling their avatar using the WASD or arrow keys to translate and rotate), they are instructed to collect firewood from the surroundings of the campfire by following the arrow, which will guide them to piles of firewood. This is encoded in the "Displacement" state of the state machine, and here again we make use of the waypoint guidance module. After the participant has collected the firewood, they are instructed to return to the (now invisible) campfire.

Before the participant is allowed to start their return, they are asked to place a marker at the location at which they suspect the campfire (during this phase the avatar's translation was disabled, but the participant could still look around). This is encoded as the "Pointing" state of the state machine, in which the Pointing and Highlighting module of the VNT is activated to enable pointing functionality.

Once the participant has placed this pointing marker (using the mouse cursor which is only visible during this phase of the trial), the marker is removed and the "Homing" state is activated in the state machine. The participant is then allowed to move towards their suspected goal. The state machine stays in this state until the participant has moved at least 12.5 meters away from the location of the first pointing when it transitions back into the "Pointing" state. Once this is complete, the participant is allowed to continue homing and the machine returns to the "Homing" state until the participant decides to end their homing attempt or a timeout condition is reached (not ending the homing attempt after 60 seconds), either of which eventually causes a transition to the "Trial End" state, which triggers a resetting of the whole system for the following trial.

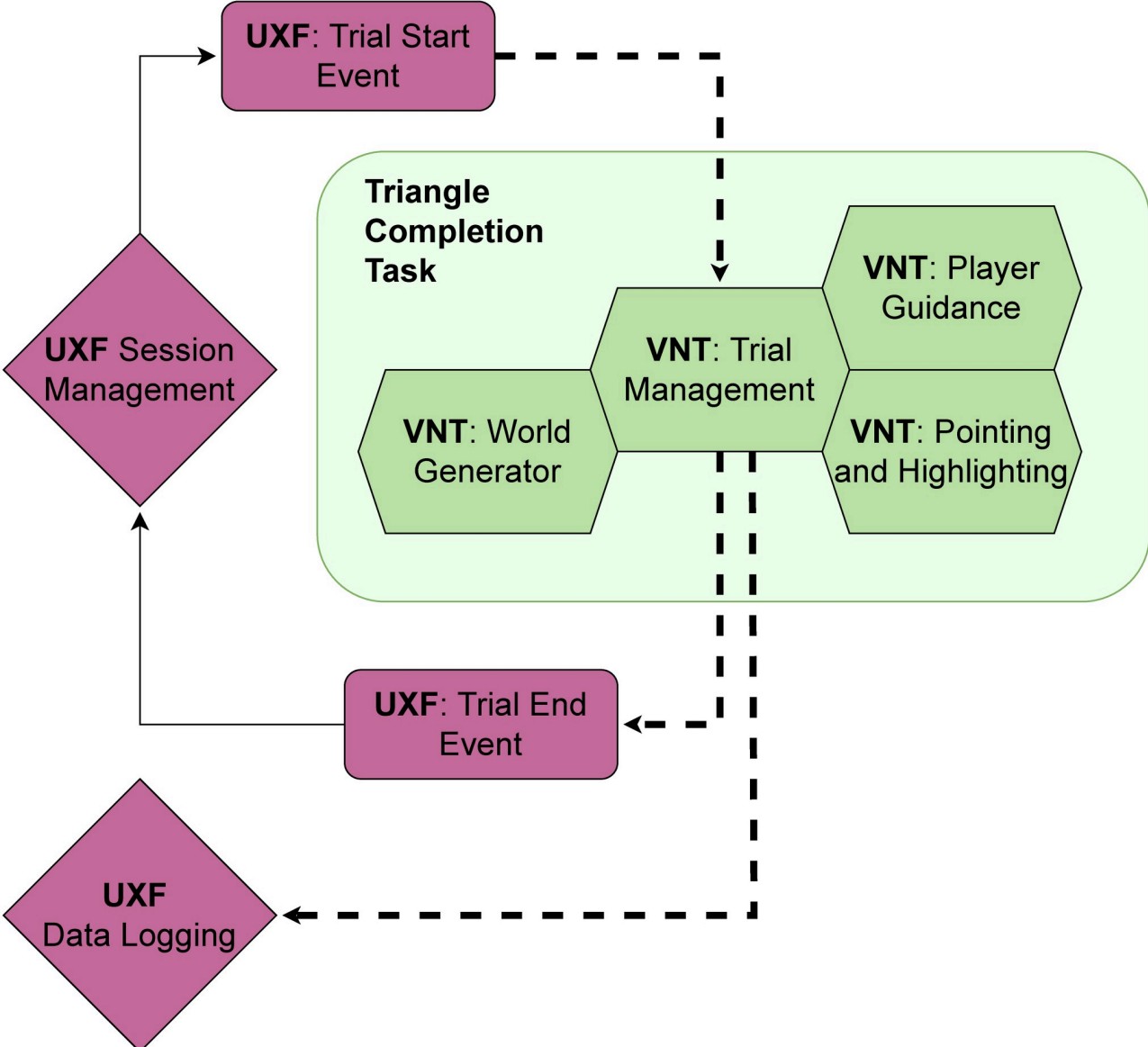

**Fig 2. Integration of the VNT with existing tools.** The modular nature of the VNT makes it easy to integrate its components with other existing tools. Here we showcase, how the implemented triangle completion task makes use of the powerful session management and data logging features of the Unity Experiment Framework (UXF) while providing the main features required for the task in question. Specifically, the UXF provides us with the basic framework of triggering a chain of trials within experimental sessions, but does not define what happens within a single trial, which is then accomplished by a combination of VNT modules, coordinated by the VNT trial state machine ("trial management"), which in turn passes the collected data of the current trial back to the UXF for saving.

## 4. Technical characterisation of the VNT

Each module in the VNT is designed to provide a specific feature set to speed up experimental design. In the following sections, we introduce the modules available in the initial version of the VNT, which have been named in the trial walk-through above. An up-to-date list of all available modules is provided in the VNT project repository (see section 'Toolbox Availability'). The project repository also features a wiki in which each module is introduced and documented in detail to help new users find a good start with the toolbox.

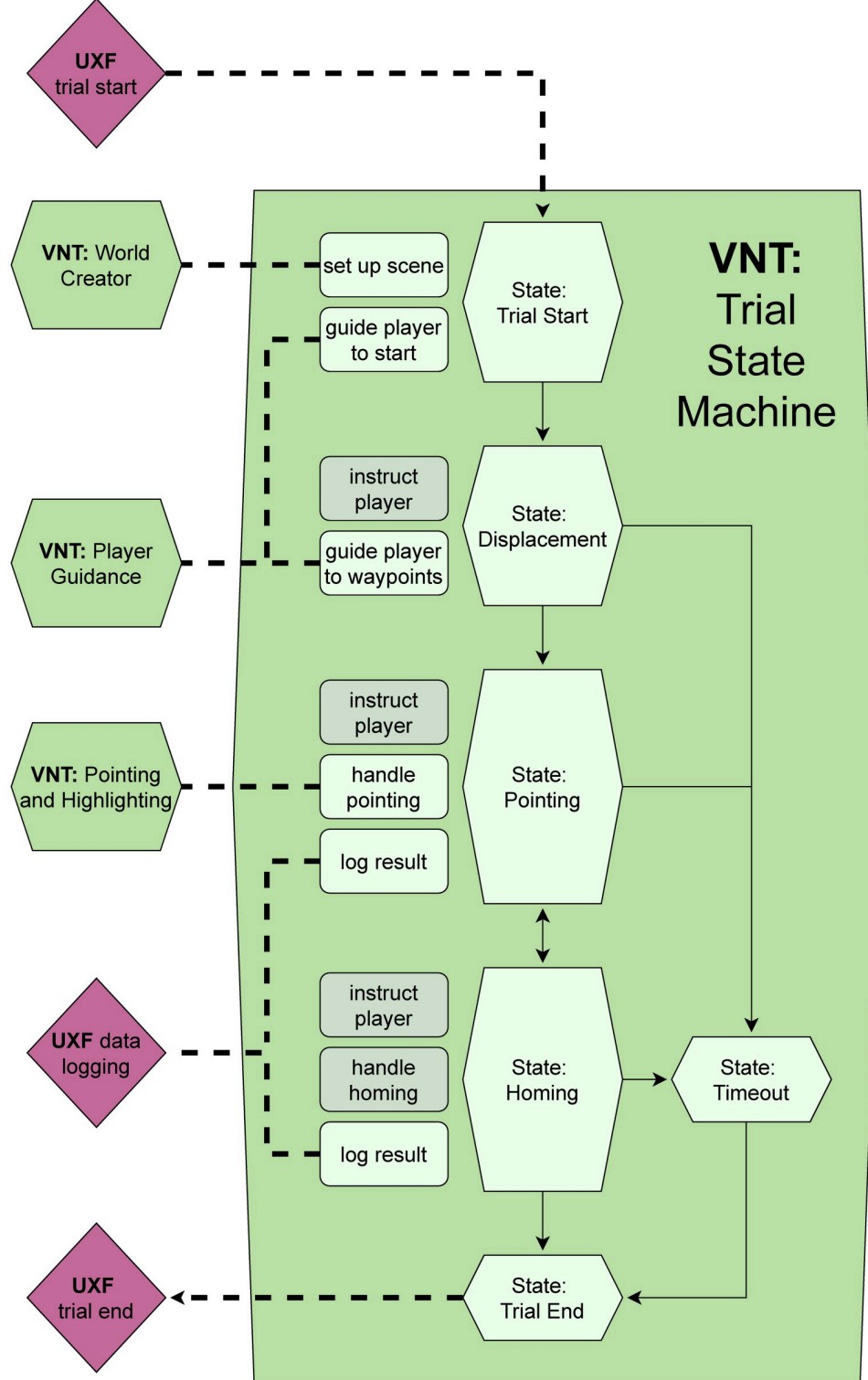

**Fig 3. Illustration of trial flow.** The flow chart visualises how the program progresses through a trial of the triangle completion task of our showcase study. Trial flow is managed by the VNT trial state machine module, which in turn calls up functions from the other VNT modules, as well as from the UXF framework. This visualisation reveals the complexity of interconnected components that need to be managed when designing VR experiments, even when only considering a relatively simple design like the triangle completion task. A state machine, as we have provided with the VNT and used for the showcase study, helps make this complexity manageable by packaging the different phases of the

experimental trial (Displacement, Homing, etc.) and clearly defining the logic that governs the transitions between the states (i.e. phases) of the experiment. Rounded boxes summarise the tasks managed within each state. Light green tasks are handled by a VNT module or the UXF, while grey tasks are currently handled in a project-specific manner, but could be included in the VNT as part of the ongoing toolbox development.

The design of VNT modules favours modularity and encapsulation of components wherever possible. This approach, based on the SOLID [36] principles of software development, is applied as far as possible in the entire toolbox. This means on a large scale that VNT modules are meant to be as interchangeable and loosely coupled as possible and on a small scale that even components within individual modules are encapsulated to maximise re-usability of code for different projects.

An example of this is the re-use of the pointing and highlighting module, which can be easily extended to process gaze data obtained by eye tracking equipment, providing a 3D gaze tracking and visualisation with object hit detection in another project (see Table 1). Such an adaptation can be achieved easily by replacing the mouse-based pointing components with gaze-based ones (see section 'Abstract classes and Interfaces' for a short summary of how the modular architecture of the VNT facilitates such changes).

## 4.1. Integration with existing tools

One popular framework for Unity-based experimental design which has been published during the last years is the Unity Experiment Framework (UXF) [21]. Among other things, the UXF provides powerful tools to structure experiments into a Trial-Block-Session structure which lends itself well to many cognitive and psychophysics experiments. These tools are integrated with useful logging features which can handle the user-defined data captured during the experiment.

In our showcase study, we use the UXF for data handling and session management, which illustrates how the VNT can be integrated with existing frameworks. However, as the session management provided by the UXF is designed to be task agnostic, it cannot be used to handle what happens within individual trials of an experiment. What at first sounds like a small coverage gap turned out to be most of the functionalities needed for our showcase study as well as VR navigation studies in general. This is illustrated in Figs 2 and 3 as well as Table 1, which show that within the bounds of the UXF framework's session management and logging features, all the actual steps of the experiment, as defined above, still need to be provided. It is also important to point out, that even though we have used the UXF for our showcase study, the VNT also provides its own data logging features (see section 'Utilities'), so users are not required to use the UXF in conjunction with the VNT.

## 4.2. Trial state machine (fulfils requirement "manage flow of trial")

To handle the flow of trials, from trial start to displacement to homing and trial end, we made use of the concept of a state machine. A state machine is an abstract computational model of a system which can be set to a finite number of different states, but can only ever be in one state at a time. As an example, consider a door: a state machine model of this door might have the states "open" and "closed". Importantly, the door can never be open and closed at the same time. The state machine consists of the definitions of each state, as well as rules which allow for transitions between them. For example, we might extend the model of the door to have additional states indicating whether it is locked or unlocked. The state machine could then be

defined so that the open state can only be reached from the unlocked state, thus making sure that only an unlocked door can actually be opened.

Our modular state machine makes use of the highly performant C# event framework to quickly communicate changes in the state of an experimental trial. In the main toolbox repository, we provide a generic state machine, which can be easily integrated into different experiments, as well as a set of example states to show how to use the provided template for more specialised needs.

For our showcase study, this means that the trial state machine can provide the control of the individual steps of a single trial of the experiment, which is a level of control more detailed than what the UXF can provide. Fig 3 shows how the logical structure of a single experimental trial was realised with the help of the state machine. For a more complete overview of the specific structure of the state machine used in the showcase study, visit the showcase repository (see section 'Showcase study repository').

## 4.3. Waypoint-based player guidance (fulfils requirement "guide player during outbound path")

In many spatial navigation experiments, participants move between several locations in an environment. Although users can be passively displaced ("teleported") to a new location in VR, it is also often meaningful to guide participants along a certain path, for example, to let them learn the visual scenery along a predefined route, provide them with PI information, or allow them to experience the environmental topology in a controlled manner [37].

We therefore created an intuitive and fully encapsulated system to provide waypoint-based guidance in virtual environments. This system allows for the dynamic creation of waypoints during experimental trials. Waypoints can be designed to have any visual representation and new waypoints can easily be created at runtime while the participant explores the virtual environment.

Waypoints automatically register and communicate when they have been reached by the participant, allowing for highly interactive experimental designs, where reaching a certain waypoint can trigger the next part of the experimental trial dynamically without having to rely on timers or other rigid predefined control measures. The set of waypoints to be used during an experimental trial can be provided by the experimenter when designing the experiment and can even be updated while the program is running.

## 4.4. Pointing and highlighting (fulfils requirement "enable pointing at goal location")

To better understand how participants solve spatial navigation tasks, it is helpful to ask them for direct input or feedback, to later examine which features or regions of the visual scene were used during the task and how this relates to accuracy and precision in navigation. Two often required modes of interaction are pointing at locations in the environment (such as at the goal or a landmark (e.g. [33])) or highlighting objects to mark them as seen or important.

We provide highly modular systems which execute the required background calculations to provide pointing and object highlighting functionalities, as well as ready-to-use sample implementations of those systems and an additional component which allows for easy logging of objects that have been interacted with. This means that experimenters can import the tools required to include pointing or highlighting in their experiment with only minimal additional integration work.

For our showcase study, we included a 3D marker placement task, for which the Pointing and Highlighting module provided the basis of implementation.

In the showcase experiment, the marker can be moved via mouse or analog stick of the controller, however other control schemes can be easily implemented by creating a new *Pointing-Controller* script, which can then be swapped in via the Unity inspector (for a more detailed description of the components of the Pointing and Highlighting module see Fig 4 and the online documentation of the toolbox linked in the 'Toolbox Availability' section).

### 4.5. Tile-based level generator (fulfils requirement "generate environment based on settings")

The level generator makes it very easy to design experiments that require different spatial environments for different experimental conditions. For example, we might need experimental arenas of different sizes with objects placed at different locations. One way of solving this problem can be to manually prepare each of the required virtual environments (often called "levels" in video games). However, this process can become very time-consuming and, depending on the complexity of the environment, also prone to errors, like accidentally displacing objects or other environmental features while editing the virtual environment.

To avoid such errors and to enable faster creation of different environments, we provide a simple level generator tool, which can be used to create environments with flat grounds and different objects placed at positions of choice. It also features randomised placement of foliage items on the ground to provide optic flow.

The level generator takes in the desired size of the environment and a list of locations at which objects are meant to be placed. The experimenter can easily define the visual appearance and placement of these objects and foliage items within the Unity editor. Furthermore, we provide functionality which dynamically extends the current environment in the direction of travel of participants as they move through the world, thus allowing for infinitely large environments without incurring the cost of pre-rendering all parts of the world even though the participants might never visit them. The Tile-based World Creator module fulfils the level generation requirement for our showcase study.

### 4.6. Utilities

We also included a collection of smaller tools that simplify various aspects of development in the Unity ecosystem. The most important of these are a highly customisable logging system and the functionality to drag-assign interface implementations via the inspector. The logging system can be used to manage logging output of other components of the toolbox and the experimenter's own code. These logging tools can take the place of the data logging functionality provided by the UXF in the showcase study (see Table 1).

The interface management feature facilitates the quick exchange of module components, like different variants of the pointing system described above. For a more detailed explanation of the value of the interface driven design employed by the VNT, we refer the reader to the section 'Abstract classes and Interfaces' below.

In addition to the logging and interface management features, a set of functions is provided, which can be used to extract any chosen files created during an experiment as a zip archive and deposit them at a destination of choice. This is especially useful for remotely deployed experiments to package result data and deliver it to the participant's desktop for further handling.

The Utilities module also contains several starter assets we have created to help other researchers quickly build working experiments using our toolbox. These include 3D models of trees, piles of wood, a campfire, rocks, a guidance arrow, and tufts of grass. All models are designed in a low-polygon style, allowing users to include many instances of these objects without incurring high rendering costs when running an experiment.

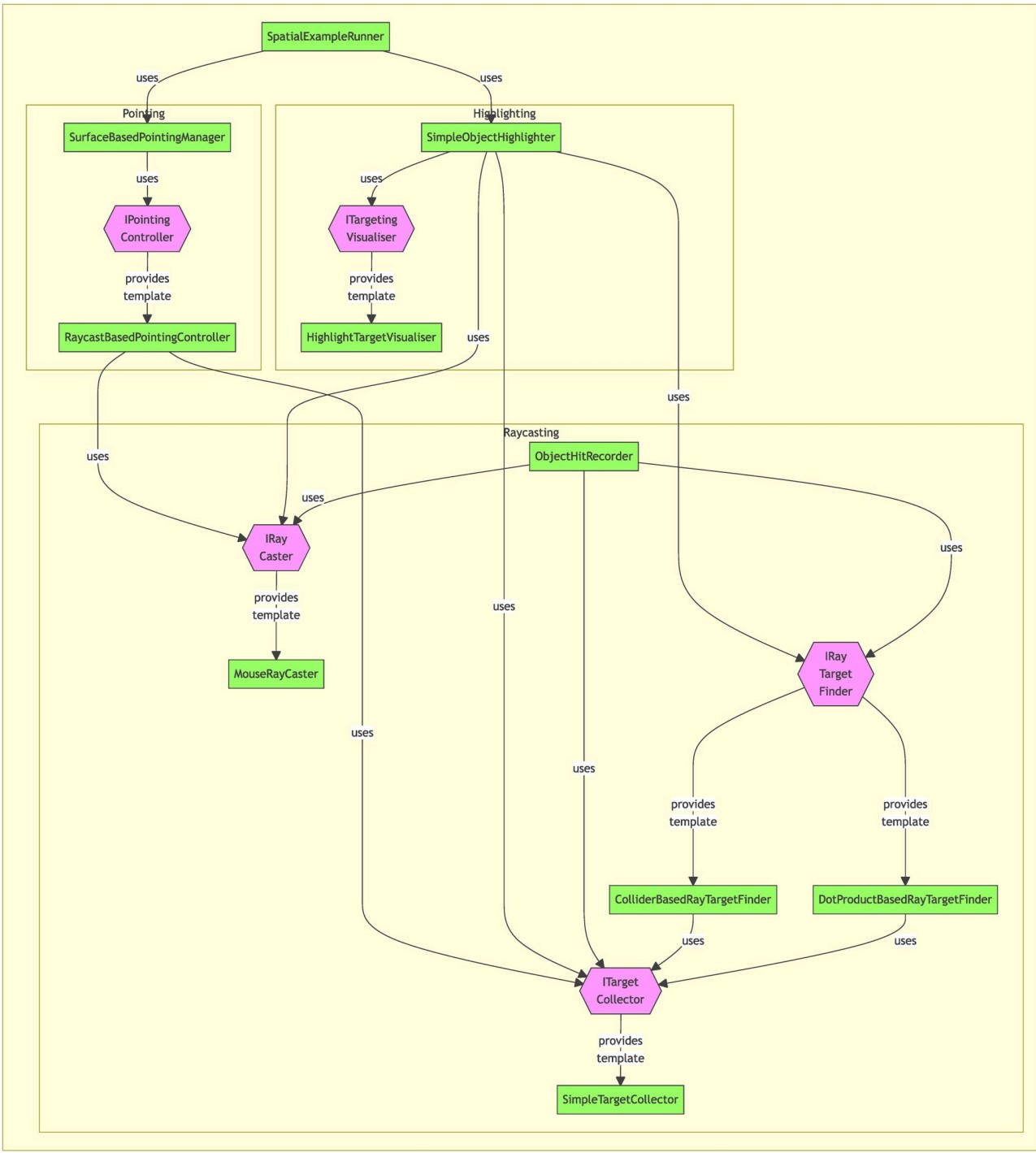

**Fig 4. Structure of the "Pointing and Highlighting" module.** The module features several components, listed below. **Pointing**: The pointing system is centred around the **IPointingController** interface, for which we provide a default implementation called RaycastBasedPointingController. The SurfaceBasedPointingManager allows the user to define a list of objects which can be pointed at (using the raycasting system, see below) and handles the actual pointing process, including placement and movement of a customisable pointing-marker-object in the 3D scene. **Highlighting**: The highlighting system features flexible components for collecting and managing possible targets, as well as doing the actual highlighting of targeted objects in the scene at runtime. This **ITargetingVisualiser** interface simply defines the functionalities needed mark a GameObject as targeted or untargeted. We provide an implementation of the Interface called HighlightTargetVisualiser, which can change the rendering material of objects to show they are being targeted. The HighlightTargetVisualiser is then used to fill the requirement for a **ITargetingVisualer** in the SimpleObjectHighlighter. **Raycasting**: The raycasting system provides the necessary functionality to do raycasting and identifying targeted objects. It is built around the IRayCaster, **ITargetCollector** and **IRayTargetFinder** interfaces. The **IRayCaster** interface simply allows the casting of rays from a specified source. We provide the MouseRayCaster as a

default implementation of this interface, which is also used in the example scene. Note that just by exchanging this component, a system using a completely different set of raycast inputs (such as an eye-tracking system) can be created and the different raycast input modes could even be exchanged on the fly in a given project. To check if a given ray has hit a target, we can use an implementation of the **IRayTargetFinder** interface. Here, we provide to implementations, which can be exchanged based on task requirements. The first implementation (DotProductBasedRayTargetFinder) determines targets based on the relative directions of the ray and the straight line between ray source and (potential) target objects. The second implementation (ColliderBasedRayTargetFinder) uses Unity's Collider system to directly detect, whether a given object was hit. Finally, to manage which objects should be targetable, we can use an implementation of the **ITargetCollector** interface. We provide a simple default implementation (SimpleTargetCollector), which can be supplied with lists of GameObjects to track as being targetable.

## 5. Designed to be changed: Using code architecture to facilitate re-use

The VNT is written with maximal modularity in mind. This is reflected on all levels of design, from whole modules to the architecture of individual scripts. Below we will briefly summarise some important technical aspects that underpin the commitment to modularity and ease of adaption of the toolbox.

### 5.1. Assemblies and namespaces

The VNT is built for the Unity game engine. Code within the ecosystem of this engine is usually written in the C# programming language. The C# language is (largely) focused on object-oriented programming. Object-oriented programming emphasises the organisation of code around objects, which are instances of classes that encapsulate data and behaviour. This promotes code re-usability, modularity and modelling of complex concepts and relations and makes object-oriented languages like C# well suited to the design of interactive software, like virtual spatial tasks.

Within the C# ecosystem, an *Assembly* is a library which holds your scripts as well as references to other required libraries. Fundamentally, it is the structure that defines a self-contained unit of functionality, as well as formalising the ways it relies on other, external pieces of code. This is especially important for larger projects, since the Unity engine needs to recompile all code every time it is changed. And since the unit of organisation is the Assembly, if all our code were collected inside one single Assembly, the entirety of our codebase would be recompiled every time we make a change to a single script. This leads to long wait times and can slow down development considerably. To prevent this problem and ensure that relationships between different functional units of code are well defined, we have encapsulated each tool in the VNT toolbox in its own Assembly.

Within each Assembly, code is organised within different *namespaces*. A namespace organises how tightly different parts of our code are linked, and which parts of our code are visible to which other parts. This means that classes from within the same namespace can easily interact with each other, while using functionalities from another namespace requires importing said namespace via the using keyword, similar to for example the **import** keyword in python). Using this principle, we have organised the code of each module of the VNT to be contained in its own namespace.

Together, these design decisions help shorten compile times during development and help keep the code organised.

### 5.2. Abstract classes and interfaces

The modules of the VNT make heavy use of two central features of the C# language called **abstract classes** and **interfaces**. Both of these language features are great ways of producing more modular and re-usable code.

In short, abstract classes can provide functionality that derived classes can implement or override. Interfaces on the other hand only define functionality, but not implement it. They can be seen as contracts which needs to be fulfilled by any class implementing the respective interface. Using this feature allows us to build systems in a way that allows easy exchange of components, without needing to rewrite code for each change.

This concept of interfaces is of special interest in the context of the Unity engine, because the C# language limits each class to be derived from only one (abstract) class, but allows it to implement many different interfaces. This is crucial, since many classes benefit from deriving from the predefined UnityEngine "MonoBehaviour" class, which allows instances of the class to be manipulated within the Unity inspector (a feature that is also central to the functioning of other packages like Landmarks or UXF). This means that in many Unity projects, most classes "are" MonoBehaviours and thus cannot inherit from another (abstract) class, blocking the use of this useful language feature. However, importantly, such classes can still implement as many interfaces as we like, which allows us to design modular and re-usable code even within the restrictions of the Unity ecosystem. To facilitate the use of interfaces, the 'Utilities' module of the VNT contains tools to hook up interface implementations within the editor, an often requested feature which is sadly still missing even in the most recent versions of the Unity engine.

## 5.3. Example of interface driven design

To provide a concrete example for the flexibility afforded to users by adapting an interface-driven design, we will briefly describe the ray-casting system, which forms part of the 'Pointing and Highlighting (fulfils requirement "Enable pointing at goal location")' module of the VNT. Ray-casting in Unity allows to check if rays, defined by an origin and a direction in the virtual space, collide with relevant objects in the scene.

The VNT provides mouse-based ray-casting, in which the position of the mouse cursor on the screen is used in conjunction with the 3D position of the virtual camera to provide pointing capabilities in the virtual world.

If we now want to replace this mouse-based system with one using gaze data provided by an eye tracker instead, this can be achieved very easily, because the module in question is designed around a series of interfaces (see Fig 4). We can simply write a new implementation of the raycasting interface (**IRaycaster**), which receives data from our eye-tracking software of choice, and plug it into the module at the same spots we would have used the **MouseRayCaster** before. This way, by changing only a single script we can immediately make use of eye-tracking in all other parts of the module, like the object highlighting, object hit detection and of course pointing systems without having to change any other part of our code-base. Following the same principle, we have provided two implementations of the **IRayTargetFinder** interface, allowing users to switch between implementations using either the dot-product of vectors or the intersection with colliders in the 3D scene to detect whether a Ray has hit a relevant target, making it possible to detect the targeting of objects even if those objects do not make use of the Unity collider component (see Fig 4).

The VNT even goes beyond this purely architectural step of defining interactions between components via interfaces. As described above, we have included the functionality to assign interface implementations via the editor, thus allowing for quick drag-and-drop exchange of interface implementations (e.g. GazeRayCaster vs MouseRayCaster) within the Unity inspector. We have illustrated this in Fig 5.

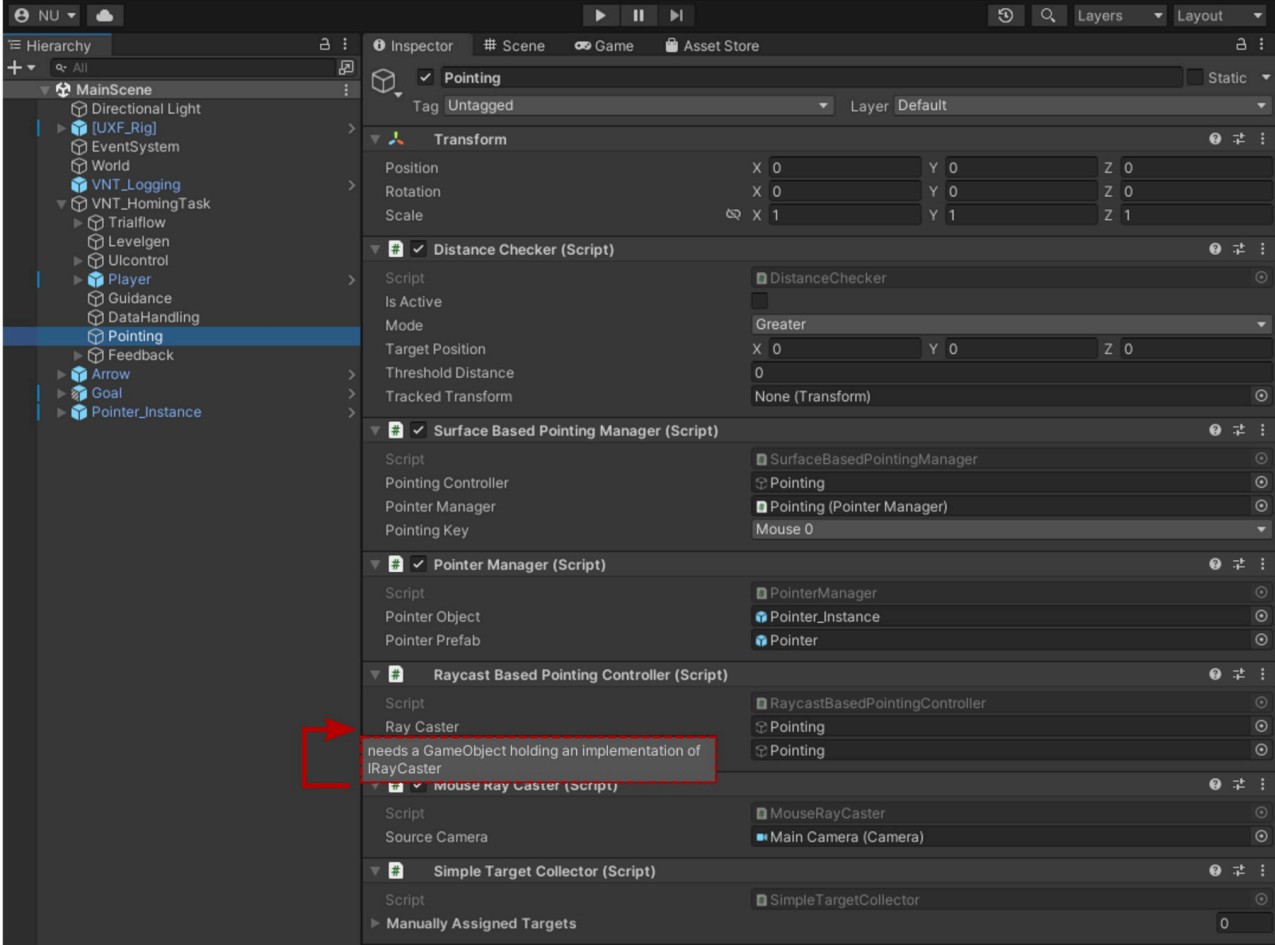

**Fig 5. Example screenshot of unity scene hierarchy and inspector.** Shown is the scene hierarchy of the main scene of the showcase study within the unity editor, together with the inspector window showing the components of the VNT pointing module in that scene. For the screenshot, the mouse was hovering over the **RayCaster** field of the **RaycastBasedPointingController** component, to show the tooltip which indicates that this field makes use of the VNT feature to drag-assign interface implementations (red dotted box and arrow). This feature exemplifies the commitment of the VNT to ease of adaptation as it enables easy swapping out of components, like replacing the **MouseRayCaster** shown here with a RayCaster based on gaze data (see example in section 'Toolbox Availability').

## 6. Evaluating the showcase study

Using the modules described above and the Unity game engine, we realised the showcase experiment to our specifications. This takes the form of a so-called "Unity project", which can be compiled into a stand-alone application that can be run on the participants' own devices. We provide the full Unity project for our showcase study as well as the corresponding pre-compiled stand-alone application (see section 'Outlook: putting the VNT to use').

Note that while technically not part of the toolbox itself, the project repository can be used as a template for building similar experiments and working with the VNT in general.

### 6.1. Results of the showcase study

In our showcase study, we recorded goal location estimates for a triangle completion task in virtual environments. From these estimates, we calculated metric position errors, i.e. how far

from the actual goal the participants placed their goal estimate derived from homing or pointing.

The triangle completion task was carried out using left- and right-handed versions of the same spatial layout (see also legend of Fig 1), however, results were combined for the summary figures shown here.

In the first part of the showcase, the virtual environment contained no landmark information and the manipulated parameter was the forward translation velocity of the participant's avatar. Results of this part of the showcase experiment did not show any difference in homing or pointing performance across different avatar velocities for the path integration task in the virtual steppe (for all speed comparisons, see Fig 6).

In the second part of the showcase study, where velocity was held constant, but one, two, or three trees were present in the different experimental conditions, homing and pointing performance seems to be affected by the number of trees available (see Fig 7). We investigate this effect of landmark count further with a more appropriate sample size in an upcoming study (Müller & Scherer et al., in prep).

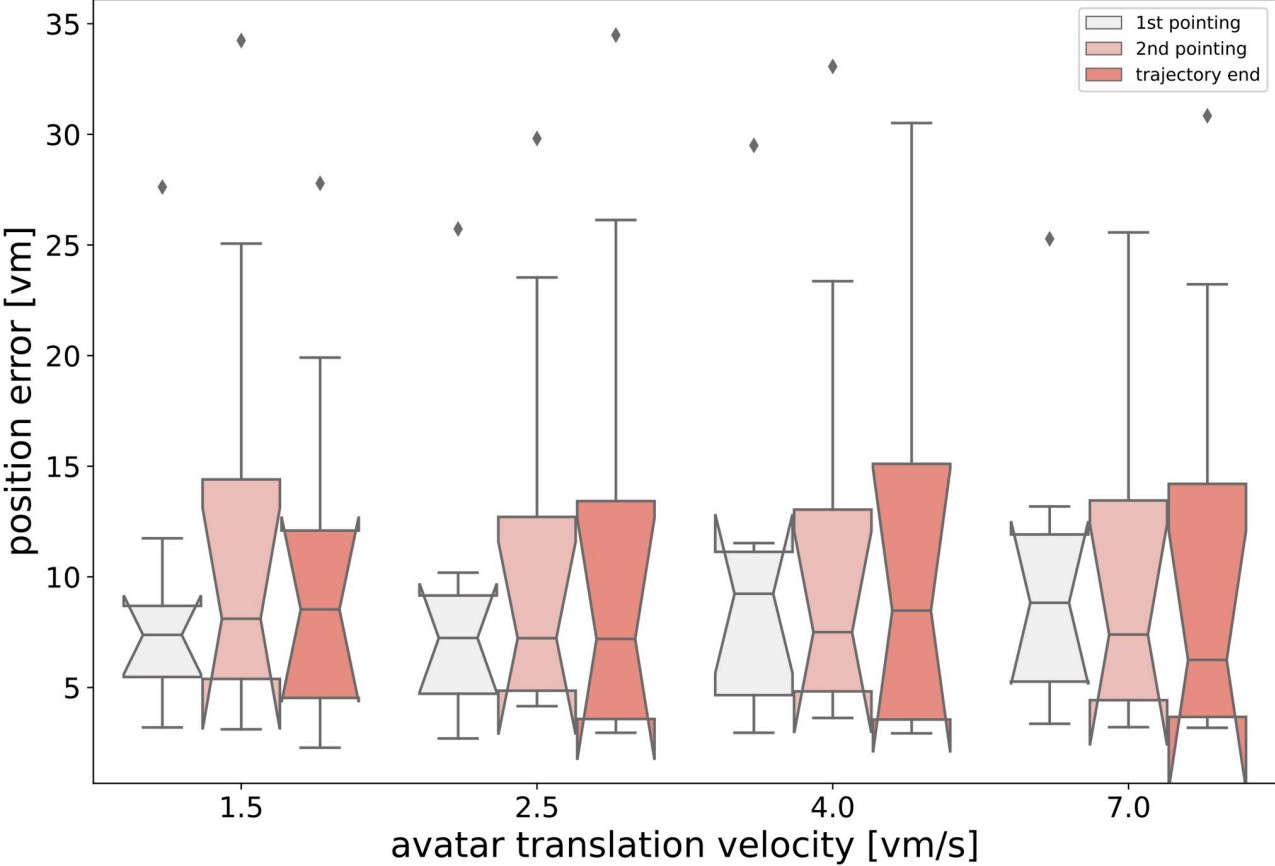

**Fig 6. No effects of avatar velocity on homing performance.** Shown are the results of an analysis of the effect of different translation velocities on performance in a triangle completion task in a virtual steppe (see 'Showcase Part A: homing at different speeds' in main text). Position errors for integrated pointing trials at the start of the return leg (1st pointing) and after a distance of 12.5 meters from the start of the return leg (2nd pointing) are shown together with the final position error of a given trial (trajectory end). Boxes show the median response for each experimental condition (from n = 17-24 repetitions) for each of N = 8 participants. Notches show 95% confidence intervals around the median, whiskers extend to 1.5x IQR, and diamonds are outliers.

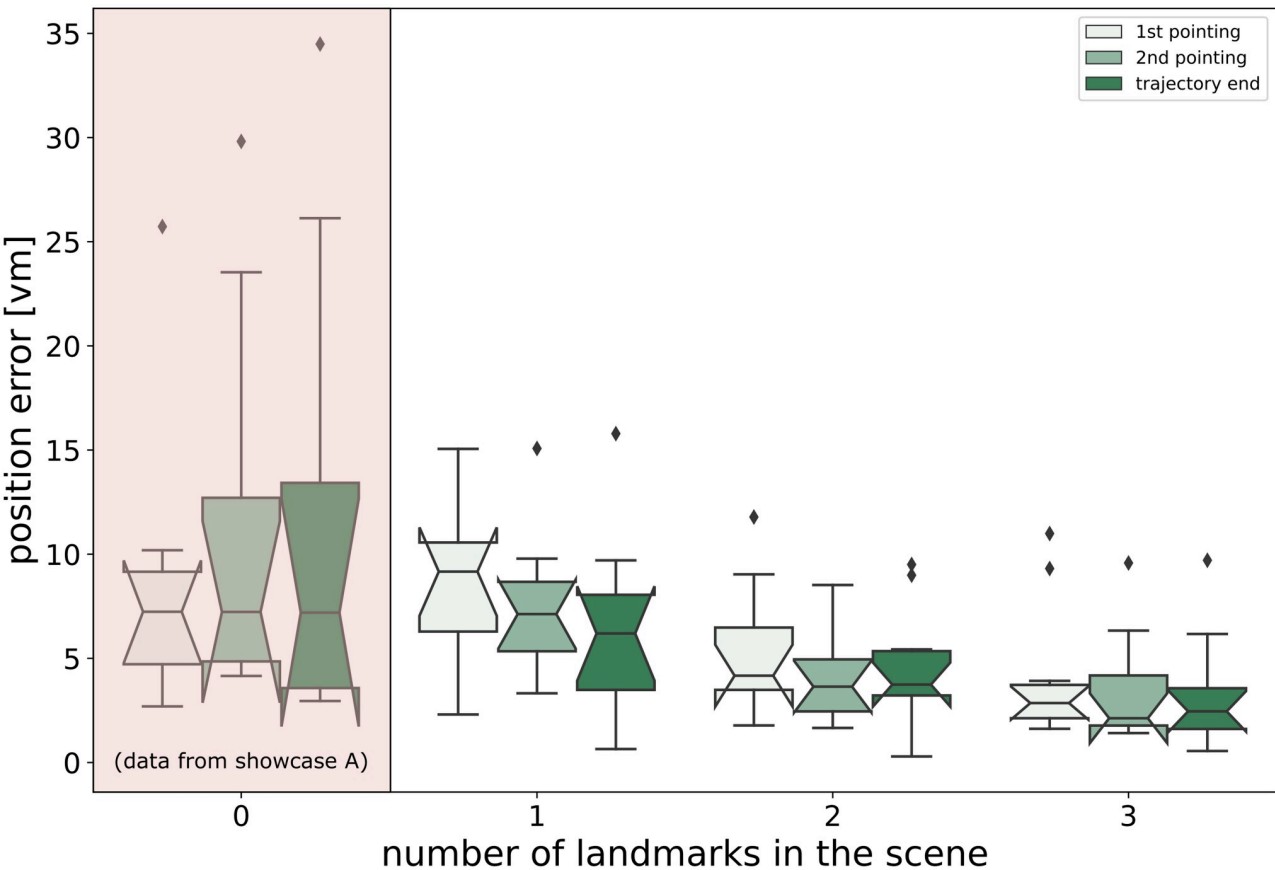

**Fig 7. Reduction of homing errors with more landmarks.** Shown are the results of an analysis of the effect of different numbers of identical landmarks (trees) on performance in a virtual triangle completion task (see 'Showcase Part B: homing with landmarks' in main text). The leftmost, separated part of the figure shows comparison data for the same translation velocity (2.5 m/s) collected in the previous experiment (Showcase A) in which no landmarks were present but the walked path was the same. The rest of the figure shows data from Showcase B. For each experimental condition (1-3 landmarks), position errors for integrated pointing trials at the start of the return leg (1st pointing) and after a distance of 12.5 meters from the start of the return leg (2nd pointing) are shown together with the final position error of a given trial (trajectory end). Boxes show the median response for each experimental condition (from n = 18-24 repetitions) for each of N = 10 participants (different group from showcase A). Notches show 95% confidence intervals around the median, whiskers extend to 1.5x IQR, and diamonds are outliers.

Results of the showcase study hint that homing performance seems to be robust across a large range of translation velocities of the participant's avatar (see Fig 6). This is reassuring, as choice of the translation velocity is one of the variables that must be considered when designing navigation tasks in VR, but is seldom the main factor of interest for the experiment.

Furthermore, homing errors seem to be greatly reduced when comparing homing under pure PI conditions (triangle completion without landmarks, see Fig 6) with homing in the same basic environment, but with added local landmarks (see Fig 7). These results are consistent with those obtained by Jetzschke et al. [31], who showed homing performance improving when using up to three identical local landmarks around the goal.

## 7. Discussion: Using the VNT to design and run virtual navigation tasks

We have introduced the VNT, an open-source, free to use software toolbox to speed up and facilitate the development of virtual navigation experiments and have highlighted how the

usefulness of the VNT results from the fundamentally modular approach that was taken in its development, which encourages code reuse and allows for a faster cycle of adapting and developing new experimental designs based on existing features.

In this initial version of the VNT, we provide tools to easily create diverse virtual environments (Tile-based Level Generator), manage the flow of experimental trials (Trial State Machine), guide participants through spatial tasks (Waypoint-based Player Guidance), enable direct interaction with objects in the environment (Pointing and Highlighting) and log and save recorded experimental data (Utilities).

Each of these tools is provided in a form that allows it to be used individually, as well as in interaction with other tools. We illustrated such an integrated use of the VNT in a showcase study. In this study, we used the VNT to successfully transpose a known experimental design to a different presentation medium, namely that of distributed desktop VR, and even expand it by including a novel 3D marker placement task.

## 7.1. Performing distributed virtual navigation experiments

Distributing experiments to participants as stand-alone applications to run on their own devices was initially thought of as a stop-gap measure to carry on data collection during the COVID-19 pandemic. However, this method of data collection proved to be highly attractive, as subject acquisition was greatly facilitated due to the high level of comfort afforded to subjects by letting them carry out the experiments at home. The power of this concept was also recently demonstrated by a large-scale spatial navigation study, which collected data of 3.9 million people via a video game app called Sea Hero Quest available online for tablets and mobile devices [38].

Since the showcase study was run on the participants own devices, it necessarily was presented via a single monitor in what is often called "desktop VR". This display mode is usually thought to be less immersive than HMD driven VR [39–43] and there is general agreement that researchers should continue to strive for a greater degree of immersion in their virtual tasks. However, there is still ongoing discussion about the differences in immersion and ease of use of different forms of virtual reality systems, with several authors pointing out that different media come with their own biases and results cannot necessarily be assumed to transfer across media [40, 43, 44], let alone from VR to real-world applications by default [30].

Some authors [42] even point out that the general trend towards using HMDs, which are usually seen as more immersive, can be problematic if experimenters use this technology in non-ambulatory settings, since the sensory conflicts evoked by conflicting "immersive" visual and idiothetic sensory input can negatively impact spatial learning. In such cases, desktop VR can be the more appropriate form of presentation and interaction due to its greater ease of use and smaller potential for sensory conflict.

In addition to the question of display medium, remote deployment of experiments can be challenging since the experimenter necessarily gives up some degree of control over the experimental procedure and physical space in which the experiment is conducted. To meet these challenges, our showcase study included a training session in which the participant was instructed on how to use the provided application and carry out the task. They were then observed for a set of training trial to ensure instructions were followed. This procedure was carried out using the screen-sharing feature of the Zoom teleconferencing service and allowed us to ensure a degree of standardisation in the execution of the task across participants.

In our study, overall homing performance seems to be similar to that observed in experiments using other methods of presentation and interaction (cf. [28]: ambulant VR or [27]: large back-projection screen) for a similar task. However, due to the small sample size of the

showcase study, no firm conclusions can be drawn on this issue. In the future, the modular nature of the VNT and cross-platform capabilities of the Unity engine will enable us—and other researchers using our tools—to easily design virtual spatial tasks and deploy them on different devices, using different forms of interaction. We have already started this process by creating an HMD version of the showcase study, so that data can be collected directly comparing performance in desktop-VR and using an HMD. The HMD version of the showcase can be found within the same git repository as the desktop version (see 'Toolbox Availability' section) and serves as an applied example for the ease of adaption that the VNT brings to experimental design.

## 7.2. Evolving the pointing task: Embedded 3D marker placement

Our showcase experiment is to our knowledge one of the first to directly compare homing and pointing performance using full 3D position estimates derived from marker placement, rather than just an angular component as is commonly derived for pointing tasks asking subjects to provide only compass-like rotational inputs (e.g. [35, 45], but cf. [46] for use of a 'virtual wand' to enable 3D-pointing in VR and [47] for a virtual building placement task).

Here we observe a general agreement between homing and pointing estimates in the different experimental conditions (see Figs 6 and 7). Thus, integrated marker placement tasks may prove to be a valuable addition to the repertoire of experimental components used to estimate the participant's navigational skills and even predict their eventual homing errors to some extent.

However, to be on more firm ground regarding such predictions, a larger-scale analysis of embedded marker placement and its effects on homing tasks would be required. Accordingly, we plan to further evaluate this experimental tool and have here provided the necessary technical implementations for others to do so as well.

## 7.3. Outlook: Putting the VNT to use

We have shown how the VNT can aid in the realisation of virtual navigation experiments and how the individual tools within the toolbox work independently as well as jointly to facilitate task implementation and data collection. In closing out this introduction to our toolbox we want to once again briefly outline how we think the VNT can be best put to use by other researchers.

Firstly, we emphasise that using the VNT requires a modest degree of familiarity with programming and a willingness to familiarise oneself with the C# programming language. We explicitly do not intend the VNT to occupy the niche of low/no-code, drag-and-drop tools. Indeed, to do so would make the toolbox rather redundant since existing packages like Landmarks [19] already do an excellent job at servicing the associated user base.

Instead we designed the VNT to best serve users who face the task of implementing virtual spatial tasks which fall outside the scope of what can be accomplished using existing tools, but who can still benefit from re-using common pieces of functionality which are shared across different tasks and designs.

We are convinced that this use case covers the situation of many researchers who need to implement a specific version of a given spatial task, while considering varied requirements, like capturing spatial data in a certain way, creating different kinds of environments and especially organising the logical flow of an experiment, regardless of the specific content of the experiment in question.

In this sense, we feel the VNT can best be considered as a companion package to the popular Unity Experiment Framework [21]. While the UXF provides a great framework within

which user can implement different sets of experiments, the VNT enables and simplifies the actual implementation by providing tools which can be re-used across different spatial tasks. We emphasize this workflow in our showcase study and visualise it in Figs 2 and 3.

The initial version of the VNT released with this publication covers a set of widely used task components, which we hope will help other researchers in the field who may struggle with building their own VR navigation experiments. However, there are still many more commonly used features which are not yet covered by the VNT.

On the one hand, we plan to steadily expand the feature set of the VNT as demands for our own experiments change and grow (work on the integration of eye tracking and virtual trap-lining is already under way, see Table 1), on the other hand, we invite other researchers to contribute their own work to the VNT. Due to its highly modular nature, the VNT may become a hub of development tools and increase sharing and reuse of code under an open-source licence across research groups, to improve transparency in method use and documentation.

## 7.4. Toolbox availability

We have provided the main toolbox in its modular form, and the Unity project implementing the showcase study in two separate, openly accessible repositories.

## 7.5. Virtual navigation toolbox repository

The Virtual Navigation Toolbox (VNT) project is openly available on GitLab at https://gitlab.ub.uni-bielefeld.de/virtual_navigation_tools/unity_vnt_main. The repository contains an up-to-date collection of all modules along with a wiki-based documentation for user convenience.

If you would like to contribute to the development of the VNT, please visit the project website and follow the contribution guidelines there.

## 7.6. Showcase study repository

Separately, on GitLab at https://gitlab.ub.uni-bielefeld.de/virtual_navigation_tools/unity_vnt_showcase_triangle_completion, we provide the showcase study's Unity project, which uses modules of the VNT repository above, and the corresponding pre-compiled stand-alone application. Additionally, the repository contains the data that support the findings of the showcase study, as well as Python scripts to recreate Figs 6 and 7.

The repository also contains a version of the showcase re-configured to work with an HTC Vive Pro HMD and the SteamVR plugin for Unity. To effect this change, we simply integrated the SteamVR plugin into the existing Unity project and made a minor adaptation concerning the relative orientation of the avatar and the attached HMD camera.

The stand-alone application of the showcase was tested on the Windows platform and limitations of the UXF used in the showcase limit compatibility with other operating systems (Linux, MacOS). However, this limitation only applies to the showcase study. The modules of the main VNT package itself are platform agnostic and to our knowledge no limitations with regard to target platforms exist.

## Acknowledgments

We want to thank Anabel Kröhnert and Ramona Lesch for collecting the data for the showcase study. Additionally, we want to thank Jack Brookes for creating the Unity Experiment Framework, which inspired us to create this toolbox. We also thank Dr. Arthur Maneuvrier and two other anonymous reviewers for their careful reading of our manuscript and their many insightful comments and suggestions.

## Author Contributions

**Conceptualization:** Martin M. Müller, Olivier J. N. Bertrand.

**Data curation:** Martin M. Müller, Patrick Unterbrink, Olivier J. N. Bertrand.

**Formal analysis:** Martin M. Müller.

**Funding acquisition:** Olivier J. N. Bertrand, Martin Egelhaaf, Norbert Boeddeker.

**Investigation:** Martin M. Müller.

**Methodology:** Martin M. Müller.

**Project administration:** Martin M. Müller, Olivier J. N. Bertrand, Martin Egelhaaf, Norbert Boeddeker.

**Resources:** Olivier J. N. Bertrand, Martin Egelhaaf, Norbert Boeddeker.

**Software:** Martin M. Müller.

**Supervision:** Martin M. Müller, Olivier J. N. Bertrand, Martin Egelhaaf, Norbert Boeddeker.

**Validation:** Martin M. Müller, Jonas Scherer, Patrick Unterbrink.

**Visualization:** Martin M. Müller, Patrick Unterbrink.

**Writing – original draft:** Martin M. Müller.

**Writing – review & editing:** Martin M. Müller, Jonas Scherer, Patrick Unterbrink, Olivier J. N. Bertrand, Martin Egelhaaf, Norbert Boeddeker.

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
