## [Decision Letter · Decision Letter 0]

7 Jul 2023

PONE-D-23-13307The Virtual Navigation Toolbox: Providing tools for virtual navigation experimentsPLOS ONE

Dear Dr. Müller,

Thank you for submitting your manuscript to PLOS ONE. After careful consideration, we feel that it has merit but does not fully meet PLOS ONE’s publication criteria as it currently stands. Therefore, we invite you to submit a revised version of the manuscript that addresses the points raised during the review process.

We look forward to receiving your revised manuscript.

Kind regards,

Antoine Coutrot

Academic Editor

PLOS ONE

Journal Requirements:

This research was supported by the DFG (Deutsche Forschungsgemeinschaft).

This work was funded by the Deutsche Forschungsgemeinschaft (DFG) grant 460373158 (https://gepris.dfg.de/gepris/projekt/460373158).

OJNB, NB and ME received the funding, MMM, JS and PU were funded as part of the project.

Reviewers' comments:

Reviewer's Responses to Questions

**Comments to the Author**

1. Is the manuscript technically sound, and do the data support the conclusions?

Reviewer #1: Partly

Reviewer #2: Yes

Reviewer #3: Partly

2. Has the statistical analysis been performed appropriately and rigorously? 

Reviewer #1: No

Reviewer #2: Yes

Reviewer #3: Yes

3. Have the authors made all data underlying the findings in their manuscript fully available?

Reviewer #1: Yes

Reviewer #2: Yes

Reviewer #3: Yes

4. Is the manuscript presented in an intelligible fashion and written in standard English?

Reviewer #1: Yes

Reviewer #2: Yes

Reviewer #3: Yes

5. Review Comments to the Author

Reviewer #1: I think that the tool proposed by the authors could be, in the future, very interesting in the context of spatial cognition. I can only agree with the principle of cooperative sharing of code and tools that could benefit the entire field. However, the presentation of the tool and the article in general seem to me a bit shaky as they are, and could benefit, in my opinion, from modifications.

1/ My main remark concerns the structure of the article. Indeed, I'm not a fan of the structure of the article proposed by the authors. In fact the article follows the traditional structure “introduction / materials and methods / results / discussion”, which seeks to resemble the format of empirical studies, but lacks the problematizations, objectives, and hypotheses but also the theoretical content to be one. I would advise to abandon this structure and do something more straightforward and more precise on the presentation of the tool according to the users' needs. For example (Bebko & Troje, 2020). Indeed, although I appreciate the reflection on the needs of researchers that led to the creation of the VNT tool, I find that the article does not really support / and or discuss the use of the tool itself. I personally imagine much more a presentation with sections driven like "Why virtual reality is of interest in the field of spatial cognition, examples" -> "How VR work in these cases and what needs for research, examples” -> “VNT proposal and presentation” -> “VNT testing through the showcase study” -> “Discussion on the use of VNT in the showcase study and succinct reporting of the study’s results” → “Global discussion on VNT”. The article already looks in many ways like this but I'll take it all the way.

2/ In addition, the place of the showcase study in the article is a bit strange at the moment: I think it should exist only to make you think about the global tool presented here: the needs, what has been allowed, how it would have been otherwise without VNT, how did VNT benefit from the study etc. In this context, the showcase study should not be, in my opinion, a scientific empirical study in the “true” sense of the word (e.g. using null hypothesis testing via statistical inferences) and should not be presented as such (I’m not sure about making statistical analyses), but a demo accompanying the presentation of the tool (with, if wanted, descriptive statistics rather than inferential). If the authors decide to keep the inferential statistical analyses, then a much more thorough presentation of the experimental method (description of the sample and the context of the test, setting of operational hypotheses, quantification and description of the variables used...) and theoretical background (What are the contributions of the study to the field of spatial cognition? Which theoretical hypothesis is tested here? What is the relation with previous works?) is needed. Indeed, the presentation of this study’s results at the same level as an abstract description of a state machine in the "results" section makes the whole thing a bit confusing and less impactful. Finally, and like many epistemologists and statisticians, I am opposed to the use of the terms "significant / non-significant" to talk about the results of the showcase study, especially considering its deep exploratory nature (Amrhein et al., 2019; Colquhoun, 2017).

3/ Another point that could be discussed is that it remains debatable for an experienced programmer that starting from scratch is sometimes more comfortable and faster than adopting and appropriating an existing code, especially in the absence of a precise and complete documentation. I think here that the authors should insist more on the proposed global "package" which allows to provide researchers with a whole set of tools which they would not necessarily have thought of but which can prove to be fundamental in the field, for example the dynamic recording of the positions and rotations of the participants (euler angles). Indeed, proposing a whole set of functions already established allows researchers to have a much wider range of functions at their disposal than what would have been built from scratch for the needs of a study, and this range could even push the innovation of measurements by allowing the exploration of other "unexpected" data.

4/ Since the tool needs to go through the Unity editor, this makes it sensitive to the version of the software used. I did not test it on all versions, but I already found incompatibility with common versions. This should probably be quickly discussed in the article.

5/ The author say that the tool is “platform agnostic”. I think this deserves more explanation. Is this the standalone build? Have the editor's tools been tested for windows mac and linux? The build for android and ios? Have this been tested on common VR systems like Oculus or HTC Vive ?

6/ The author describe, for example in the abstract, the use of “virtual reality which allow for the creation of highly flexible and immersive study environments”. I would not call the environment produced and used by the showcase study on a computer "immersive and flexible", and I would not even give it the name "virtual reality" Would these tools work in virtual reality for example via the use of the usual plug-ins like OpenVR SteamVR etc? VR is very promising in spatial cognition and I think any package oriented toward virtual navigation should also aim for virtual reality systems and not only personal computers. This deserves much more attention, I think, in the authors' article, as I would consider adding some discussion and references on why virtual reality is used in spatial cognition, including the ecological dimension it allows. For example (Cogné et al., 2017; Maneuvrier et al., 2020; Parsons, 2015)

7/ The authors say they conducted a remote evaluation because of the Covid outbreak but that it may prove interesting in the future. Remote evaluation poses many methodological problems that are not mentioned at all in this article. This remote assessment also offers many promising avenues for diagnosis and rehabilitation, which are not mentioned any more than the problems. I think this deserves more attention as well, especially on the integration of tools such as VNT in this framework.

8/ In the showcase study, the triangle completion task is implemented and used but not really described. I would consider adding a word about the triangle completion task, its applications and uses and / or other virtual implementation of the task (Cherep et al., 2020, 2023; Dorado et al., 2019, 2019; McLaren et al., 2022; Péruch et al., 1997; Riecke et al., 2002). I would also consider describing in more details (with references) the traditional tasks covered by the tool.

Amrhein, V., Greenland, S., & McShane, B. (2019). Scientists rise up against statistical significance. Nature, 567(7748), 305‑307. https://doi.org/10.1038/d41586-019-00857-9

Bebko, A. O., & Troje, N. F. (2020). bmlTUX : Design and Control of Experiments in Virtual Reality and Beyond. i-Perception, 11(4), 2041669520938400. https://doi.org/10.1177/2041669520938400

Borgnis, F., Baglio, F., Pedroli, E., Rossetto, F., Uccellatore, L., Oliveira, J., Riva, G., & Cipresso, P. (2022). Available Virtual Reality-Based Tools for Executive Functions : A Systematic Review. Frontiers in Psychology, 13. https://doi.org/10.3389/fpsyg.2022.833136

Cherep, L. A., Kelly, J. W., Miller, A., Lim, A. F., & Gilbert, S. B. (2023). Individual differences in teleporting through virtual environments. Journal of Experimental Psychology: Applied, 29, 111‑123. https://doi.org/10.1037/xap0000396

Cherep, L. A., Lim, A. F., Kelly, J. W., Acharya, D., Velasco, A., Bustamante, E., Ostrander, A. G., & Gilbert, S. B. (2020). Spatial cognitive implications of teleporting through virtual environments. Journal of Experimental Psychology: Applied, 26, 480‑492. https://doi.org/10.1037/xap0000263

Cipresso, P., Giglioli, I. A. C., Raya, M. A., & Riva, G. (2018). The Past, Present, and Future of Virtual and Augmented Reality Research : A Network and Cluster Analysis of the Literature. Frontiers in Psychology, 9, 2086. https://doi.org/10.3389/fpsyg.2018.02086

Cogné, M., Taillade, M., N’Kaoua, B., Tarruella, A., Klinger, E., Larrue, F., Sauzéon, H., Joseph, P.-A., & Sorita, E. (2017). The contribution of virtual reality to the diagnosis of spatial navigation disorders and to the study of the role of navigational aids : A systematic literature review. Annals of Physical and Rehabilitation Medicine, 60(3), 164‑176. https://doi.org/10.1016/j.rehab.2015.12.004

Colquhoun, D. (2017). The reproducibility of research and the misinterpretation of p-values. Royal Society Open Science, 4(12), 171085. https://doi.org/10.1098/rsos.171085

Dorado, J., Figueroa, P., Chardonnet, J.-R., Merienne, F., & Hernández, T. (2019). Homing by triangle completion in consumer-oriented virtual reality environments. 2019 IEEE Conference on Virtual Reality and 3D User Interfaces (VR), 1652‑1657. https://doi.org/10.1109/VR.2019.8798059

Maneuvrier, A., Decker, L. M., Ceyte, H., Fleury, P., & Renaud, P. (2020). Presence promotes performance on a virtual spatial cognition task : Impact of human factors on virtual reality assessment. Frontiers in Virtual Reality, 1. https://doi.org/10.3389/frvir.2020.571713

McLaren, R., Chaudhary, S., Rashid, U., Ravindran, S., & Taylor, D. (2022). Reliability of the triangle completion test in the real-world and in virtual reality. Frontiers in Human Neuroscience, 16. https://www.frontiersin.org/articles/10.3389/fnhum.2022.945953

Parsons, T. D. (2015). Virtual Reality for Enhanced Ecological Validity and Experimental Control in the Clinical, Affective and Social Neurosciences. Frontiers in Human Neuroscience, 9. https://doi.org/10.3389/fnhum.2015.00660

Péruch, P., May, M., & Wartenberg, F. (1997). Homing in Virtual Environments : Effects of Field of View and Path Layout. Perception, 26(3), 301‑311. https://doi.org/10.1068/p260301

Riecke, B. E., Veen, H. A. H. C. van, & Bülthoff, H. H. (2002). Visual Homing Is Possible Without Landmarks : A Path Integration Study in Virtual Reality. Presence: Teleoperators and Virtual Environments, 11(5), 443‑473. https://doi.org/10.1162/105474602320935810

Riva, G. (2022). Virtual Reality in Clinical Psychology. Comprehensive Clinical Psychology, 91‑105. https://doi.org/10.1016/B978-0-12-818697-8.00006-6

Reviewer #2: The authors present an open source software toolbox to implement spatial navigation experiments using the Unity 3D engine. The manuscript first describes the requirements analysis, design decisions, and structure of the presented toolbox. The authors then present a validation experiment built using their toolbox and reimplementing a well-known spatial navigation task (triangle completion) with the additional component of a spatial pointing task. The manuscript is accompanied by two git repositories, one each for the toolbox and case study.

The software is novel, and the authors validate the tool on a well known task (fitting PLOS guidelines). The manuscript is clearly written and easy to follow, and the design decisions behind the toolbox become clear. I was especially happy to see that the authors focused on interoperability with existing tools (e.g., UXF), and that they include a full example study with additional documentation. However, some details about the case study were not fully clear to me in the current version, and I have included some other suggestions that might strengthen the manuscript below:

- I realized only quite late in reading that the showcase study was desktop VR-based rather than run in an HMD (most recent studies seem to equate VR == headset). The authors might want to explicitly point this out earlier in the manuscript.

- It would also be helpful to understand what parts of the toolbox apply to virtual environments in general (i.e., could also work in immersive VR), and which are desktop-specific.

- There are some details that would help the reader better understand the exact procedure of the showcase study (some of which might be in the wiki, but would be good to also detail in the manuscript):

- How was the pointing task accomplished, i.e. how do participants "place a marker" (mouse cursor, aiming the viewport, ...)?

- The showcase Results section should state more clearly what statistical tests were performed for which experiment, and what the numerical results were:

- For example, the text mentions "Kruskal–Wallis one-way analysis of variance across all measures and conditions". Since velocity was the main IV, which of the pointing or endpoint errors was analyzed here? Alternatively, if this means a test was performed for each error metric, please be more explicit about this and provide the results for each test.

- The Fig. 4 caption test instead seems to suggest that error measures were compared within each avatar velocity? However, the numbers suggest that this is the same test as above. Please clarify.

- "p ≤ 0.88126" might be a typo and indicates to p >= ?

- In general, most statistical values seem to be written as "approximately equal", which seems unusual notation to me

- The section describing participants and ethics approval on p. 9 does not mention any number of participants - please add.

- The case study wiki provides a lot of additional helpful information, some of which might be useful to include in the manuscript for readers still deciding whether this toolbox might work for their application. For example, a screenshot of the Unity scene hierarchy can give a person experienced with Unity a quick overview of the structure and objects/prefabs used.

- In my opinion, the packages in Table 1 do not actually benefit from being presented as a table - I think the details in the table caption and references would combine to make a good descriptive paragraph in the manuscript.

- I suggest naming the type of license explicitly in the text rather than just saying "open source" (I believe the abstract mentions it is CC)

- Since the authors mention a work-in-progress "trap-lining" study, it could be helpful to briefly mention what this means

Reviewer #3: In their manuscript, “The Virtual Navigation Toolbox: Providing tools for virtual navigation experiments,” Martin M. Müller, Jonas Scherer, Patrick Unterbrink, Olivier J.N. Bertrand, Martin Egelhaaf, and Norbert Boeddeker present a new Unity 3D toolbox for designing spatial navigation experiments. The authors do a noteworthy job of putting together a new toolbox that is full of useful tasks, including a waypoint-based guidance task, a triangle completion task, and a novel pointing/marker placement task. Moreover, the authors describe data from two preliminary studies. First, they found a null difference in performance as a function of translation speed in an optic-flow-based triangle completion task. Second, they found that stable landmarks enhanced performance of the triangle completion task, thus replicating previous research. I commend the authors for their hard work on this project, and I have several questions and comments for the authors, which I hope will help strengthen the impact of their manuscript.

Major comments:

1. Again, I thank the authors for their desire to make new tasks available for the spatial navigation community. However, I felt that there were several aspects of the manuscript that could have been clearer. For example, the manuscript felt to me like it was neither a comprehensive guide for their toolbox (i.e., the documentation herein was relatively sparse) nor did it feel like a full data paper (i.e., their sample sizes were very small: N=8 for Showcase A and N=10 for Showcase B). Moreover, I felt that the overall scope of the discussion was a bit narrow, and I think the manuscript would have a stronger impact if the authors included more citations, background, rationale, etc. Therefore, I recommend that the authors consider either adding significantly more information about how to use their toolbox (i.e., a sort of guide introducing their software) and/or that they significantly increase the number of participants in their sample (i.e., a data paper in which they also introduce the overall functionality of the toolbox). As it stands, I personally feel that it is a bit too brief on both the background and the data side.

2. I personally felt that the authors’ discussion of extant packages for spatial navigation was a bit misleading and inaccurate. For example, I will focus my discussion to Landmarks (i.e., their citation [30]) because I am very familiar with this package:

a. Lines 30-37: I disagree that users would have to “reinvent the wheel” in Landmarks. In fact, the key purpose of Landmarks is to allow users to generate commonly used spatial navigation experiments with VR (e.g., navigation tasks, several pointing tasks, map-drawing tasks). Also, yes, users can employ VR headsets but Landmarks makes it as simple as a click of a button in a dropdown menu to change between VR headsets and desktop versions of the task.

b. Lines 50-53: Again, I feel that Landmarks implements a very modular approach as well and all tasks/code/etc. can be recycled depending on the users’ needs.

c. Lines 318-320: note that Landmarks can perform this functionality as well.

d. Lines 322-323: again, this sounds exactly like Landmarks.

e. Lines 325-328: This sounds like really great functionality (and, as far as I know, is not currently implemented in Landmarks).

3. I feel that the authors use the term virtual reality (VR) a bit too broadly. For example, given the rise in highly immersive VR devices such as fully mobile VR headsets, I think it would help if the authors clarify that they are really talking about desktop-based navigation (i.e., to make it clear that this is a specific use case that does not include VR headsets).

4. Relatedly, given the focus on desktop-based navigation, I think that it could severely limit the impact of their toolbox. Specifically, I apologize if I missed something, but it appears to me that the main task is a triangle completion task, which would allow researchers to study the role of optic flow, etc., on the path integration system. Therefore, I feel that the toolbox is limited in scope; e.g., given that other systems such as body-based cues should likely contribute to this process (e.g., as they cite in the papers from Harootonian et al. [5, 23]). However, I also understand that there were other aspects that users could employ for different types of tasks (e.g., the “waypoint-based guidance task” and the “pointing task” seemed like especially useful tasks).

5. Related to my previous point, I might recommend that you are careful in referring to it as “path integration” because this appears to be a fully desktop-based task (i.e., only visual cues, no body-based cues). Do you envision that your toolbox will eventually be a fully VR-based task (i.e., with headset)? Either way, I recommend that you clarify these points.

6. Lines 229-230 and 305: The authors discuss the idea that users could make new tasks and functionality with “minimal integration work.” How much work? Also, what would be the nature of such work (e.g., coding in C#)?

7. Similarly, given that maintaining software packages requires a lot of effort, resources, and personnel, what plans/frameworks do you have in place to ensure the longevity of this package? (I see that you discuss that the general community can contribute, but how likely do you think this would be to occur?)

8. I feel that the authors are selling themselves short by referring to their task as a “pointing task”. Generally, we tend to refer to pointing tasks as those in which participants point in a given direction. Instead, I might recommend that the authors refer to this as a “marker placement task” or something of that nature to make it clearer that this is a novel task and that it can offer additional information (i.e., about direction and distance simultaneously). Also, I felt that the authors could add a more complete description of their task.

9. In general, I felt that the Methods section could be more detailed to explain both the tasks and the participants. For example, will you please clarify the following points?

a. The authors only mention the sample sizes in the figure captions. Moreover, why did the number of participants differ between Showcase A and B (were these separate groups of participants)?

b. Also, I was unclear on the description of the number of trials in the task (i.e., based on the methods). At first, it sounded like there were only two trials, but then the figure captions (e.g., for Figure 4 indicates 17-24 repetitions) and results listed a different number of trials. How many trials did the participants complete (and why were there differing trial counts between participants)?

10. As far as the data in this paper go, I again feel that the sample size is quite small (N=8 for Showcase A and N=10 for Showcase B), which I think is a limitation. For example:

a. Lines 364-369: the null result here might not justify combining results due to the extremely small sample size and extremely small trial count. Also, the p > 0.1 might actually be trending toward a difference? Were the trial orders counterbalanced? Was there any sign of an effect in which participants improved on the second trial (or subsequent trials; again, sorry but I am unclear on the trial structure here [e.g., what do you mean by, “or in a design which mirrored the spatial layout along the world y-axis, yielding trials with either left or right-handed turns, which were presented in randomised order,” in the caption of Figure 1]?

b. For the null result of translation velocity:

i. Should you consider a Bayes null analysis here (i.e., to determine whether the evidence is in favor of the null)?

ii. Again, the small sample size is a concern for me, especially when you are interpreting null effects (i.e., could this simply be a power issue?). Therefore, at a minimum, I would recommend tempering your conclusions here (but as I mention above, I recommend adding more participants to your sample).

11. For the landmark part of discussion, etc., I feel that you could cite more papers. Again, I feel that you could generally supplement your manuscript throughout to make it more comprehensive (including more citations).

Minor comments:

12. Lines 206-208: I was initially a bit unclear about the nature of the task. Here, I surmised that they performed the “pointing task” twice, but the description here was a bit unclear for me.

13. Line 210: what were the parameters for the “timeout” state?

14. Line 354: quotation ends funny (i.e., should be right after “Data Availability” to close quotes).

6. PLOS authors have the option to publish the peer review history of their article (what does this mean?). If published, this will include your full peer review and any attached files.

Reviewer #1: **Yes: **Arthur Maneuvrier

Reviewer #2: No

Reviewer #3: No

---

## [Author Response · Author response to Decision Letter 0]

21 Aug 2023

We are happy that all reviewers see the potential of our toolbox to aid in the development of VR-driven spatial cognition experiments and we thank the reviewers for their detailed and constructive feedback to improve our manuscript.

In collating the comments from the different reviews, we have taken as the central point of all reviews the observation that the initially submitted manuscript sat in an awkward intermediate spot in between a full introduction of our toolbox and a traditional empirical study.

We agree with this assessment and have restructured the entire manuscript to fully conform to the format of a method showcase.

This necessarily required several larger changes to the text, however, we feel these changes bring the manuscript more in line with our intention to present a comprehensive introduction to our toolbox.

We go over the different changes made to the manuscript in the enclosed response letter to the reviewers, where we respond to the reviewers' individual comments.

Furthermore, the decision letter stated:

Response: We re-uploaded all figures after processing via PACE to comply with PLOS ONE's style requirements.

This research was supported by the DFG (Deutsche Forschungsgemeinschaft).

This work was funded by the Deutsche Forschungsgemeinschaft (DFG) grant 460373158 (https://gepris.dfg.de/gepris/projekt/460373158).

OJNB, NB and ME received the funding, MMM, JS and PU were funded as part of the project.

Response: 

As requested, we removed all funding information from the Acknowledgments section of the manuscript also included the (unchanged) wording of the funding statement in the cover letter.

---

## [Decision Letter · Decision Letter 1]

16 Oct 2023

The Virtual Navigation Toolbox: Providing tools for virtual navigation experiments

PONE-D-23-13307R1

Dear Dr. Müller,

We’re pleased to inform you that your manuscript has been judged scientifically suitable for publication and will be formally accepted for publication once it meets all outstanding technical requirements.

Kind regards,

Antoine Coutrot

Academic Editor

PLOS ONE

Reviewers' comments:

Reviewer's Responses to Questions

**Comments to the Author**

1. If the authors have adequately addressed your comments raised in a previous round of review and you feel that this manuscript is now acceptable for publication, you may indicate that here to bypass the “Comments to the Author” section, enter your conflict of interest statement in the “Confidential to Editor” section, and submit your "Accept" recommendation.

Reviewer #1: All comments have been addressed

Reviewer #2: All comments have been addressed

Reviewer #3: (No Response)

2. Is the manuscript technically sound, and do the data support the conclusions?

Reviewer #1: Yes

Reviewer #2: Yes

Reviewer #3: Yes

3. Has the statistical analysis been performed appropriately and rigorously? 

Reviewer #1: N/A

Reviewer #2: N/A

Reviewer #3: N/A

4. Have the authors made all data underlying the findings in their manuscript fully available?

Reviewer #1: Yes

Reviewer #2: Yes

Reviewer #3: Yes

5. Is the manuscript presented in an intelligible fashion and written in standard English?

Reviewer #1: Yes

Reviewer #2: Yes

Reviewer #3: Yes

6. Review Comments to the Author

Reviewer #1: Dear authors and editors,

I think this version of the article, and in particular its structure, is much more relevant to the intended purpose. Overall, I find it more coherent and linear, without the showcase experimental study "interfering" with the reasoning and presentation of the software. The vast majority of my comments have been incorporated in a way that seems judicious to me, and I personally find this version more accomplished. I can only hope that the authors share this opinion. In any case, I consider the article in its present form to be ready for publication.

I'd like to thank the authors for their work and for making the tools available. Finally, I look forward to hearing more about the VNT in the coming months and years, as a tool of this kind can only be truly useful if it is developed and updated over the long term, so that a community of users can be built up, which will ultimately help to enrich it.

Sincerely

Reviewer #2: (No Response)

Reviewer #3: The authors have substantially revised their manuscript following comments from the reviewers. I feel that the authors are to be commended for addressing the concerns. I really appreciate the new additions to the paper and I think the paper will now have a greater impact for researchers in the field. I think that the authors’ decision to focus on the description of their toolbox (rather than the description of the “showcase” study) helps the paper have more focus and it now clearly reads as a description and example use-case of their toolbox. Accordingly, I especially appreciated the addition of Section 5 of the manuscript as well as the changes in the Discussion section. I also think that it is great that they added VR functionality via the HTC Vive, which I think will strengthen the number of researchers that will use the toolbox.

I have a few minor comments and suggestions that I hope will strengthen the final paper.

1) While I think the authors have done a great job of adding more details about their toolbox, I still wonder whether it might make sense to include an example script or something to provide more specific details about how users would implement new tasks within your framework. I feel that this could increase the possible user base of your toolbox and it would further solidify your paper’s role in serving as a bit of a tutorial on how to use your toolbox. I will leave this as a possible addition that should be implemented at the discretion of the Editor and the authors.

2) While I initially stated that the paper felt like neither a complete description of their package nor a complete study and I realize that this feedback is inconsistent with Reviewer 1’s suggestions, I feel that the authors should have left the statistics for their “showcase” study here (i.e., in the version of the manuscript with track changes, I feel they should not delete lines 679-680 or 686-689). However, I think that having the associated caveats that you have now added is very beneficial and I agree with the decision to pare down on the analyses, etc. for the purpose of this paper. At the end of the day, I realize that it will be up to the authors and the editor to decide whether it is appropriate to include the statistics here.

3) In the initial review the other reviewers and I raised concerns regarding the use of the term VR. Here, I feel that the authors took some steps toward fixing this concern; however, in other ways, I feel that the authors took a step even further in referring to desktop-based tasks as VR. Personally, I strongly encourage them to use the term “virtual environments” and to reserve the term VR for fully immersive tasks (i.e., with a HMD). At the very least, I think it would be beneficial to at least consistently use the term “desktop VR” whenever they are describing tasks that employ desktop-based virtual environments.

4) I found the figures to be relatively fuzzy (e.g., especially Figure 4, which was virtually illegible for me). I understand that this may have been an issue with the actual journal submission rather that the figures themselves.

7. PLOS authors have the option to publish the peer review history of their article (what does this mean?). If published, this will include your full peer review and any attached files.

Reviewer #1: **Yes: **Arthur Maneuvrier

Reviewer #2: No

Reviewer #3: No

---

## [Editor Report · Acceptance letter]

31 Oct 2023

PONE-D-23-13307R1 

The Virtual Navigation Toolbox: Providing tools for virtual navigation experiments 

Dear Dr. Müller:

I'm pleased to inform you that your manuscript has been deemed suitable for publication in PLOS ONE. Congratulations! Your manuscript is now with our production department. 

Kind regards, 

on behalf of

Dr. Antoine Coutrot 

Academic Editor

PLOS ONE